# Knowledge and Practices among Dental Practitioners Regarding the Use of Antimicrobials for Periodontal Diseases: An Online Survey in Saudi Arabia

Mohammed Sultan Al-Ak'hali [1,*], Esam Halboub [2], Mona Awad Kamil [1], Wafa Hassan Alaajam [3], Abdulaziz Mahnashi [4], Jabbar Khubrani [4], Abdullah Mahnashi [4], Khalid Mahnashi [4] and Nuha Farea [5]

[1] Department of Preventive Dental Sciences, College of Dentistry, Jazan University, Jazan 45142, Saudi Arabia
[2] Department of Maxillofacial Surgery and Diagnostic Sciences, College of Dentistry, Jazan University, Jazan 45142, Saudi Arabia
[3] Restorative Dental Science Department, College of Dentistry, King Khalid University, Abha 62529, Saudi Arabia
[4] College of Dentistry, Jazan University, Jazan 45142, Saudi Arabia
[5] College of Dentistry, University of Science and Technology, Sana'a 1247, Yemen
* Correspondence: sultanperiodontics@gmail.com; Tel.: +96-65-6983-6675

**Abstract:** This study aimed to investigate the knowledge and practices among dental practitioners in Saudi Arabia regarding the use of antimicrobials for periodontal diseases. An online questionnaire was sent to senior dental students and dental practitioners including interns, general dental practitioners (GDP), and periodontists in Saudi Arabia. Two hundred and twenty-three dental practitioners responded and participated in the study. The potential associations between the use of antimicrobials and different variables were assessed by a chi-square test. The majority of the participants (84.3%) reported prescribing systemic antimicrobials for a periodontal abscess or acute necrotizing periodontal disease. Surprisingly, 31% of participants reported prescribing systemic antimicrobials for deep localized periodontal pockets or for acute gingivitis associated with herpes simplex in children. Noteworthy is that 66% of the participants thought that mechanical periodontal treatment alone, without adjunctive antimicrobial therapy, is adequate to resolve the clinical condition in most cases of periodontal diseases. Almost half of the participants recommended the use of local antimicrobials for a periodontal pocket (45.3%), a recurrent periodontal pocket (45.4%), and refractory periodontitis (43.7%). The barriers against the use of local antimicrobials were a lack of knowledge and a lack of continuous education after graduation, as reported by 64% of the participants. In conclusion, knowledge and practices regarding antimicrobial use for periodontal diseases were inadequate, especially among practitioners other than periodontists.

**Keywords:** antimicrobial therapy; periodontal disease; knowledge; practice; dental practitioners; Saudi Arabia

## 1. Introduction

Given the poly-microbial nature of dental infections, antimicrobials are one of the most commonly prescribed drugs in dentistry [1,2]. Prescribing antibiotics for periodontal infection is limited to a very few conditions, such as for patients with aggressive forms of periodontitis, patients who are also suffering from underlying medical problems, and patients with refractory periodontitis or acute or severe periodontal infections, and has to follow recommended guidelines [3].

In their daily practice, dental practitioners regularly prescribe antimicrobials for therapeutic or prophylactic purposes, to manage or prevent oral and dental infections [4]. However, antimicrobial use is a double-edged sword. It is well-known that improper prescriptions for patients, unnecessary prescriptions for healthy subjects aiming at preventing

infections and complications, or improper use by the patients themselves are major factors in the development of antimicrobial resistance. The latter is considered an ongoing and serious global challenge, since the initial discovery of antimicrobial agents, as it affects morbidity and mortality and increases health costs [4–8].

In the context of periodontal diseases, systemic antimicrobials are indicated for acute periodontal infections where local spread or a systemic complication has occurred [9,10]. In addition, antimicrobials use as prophylaxis prior to periodontal surgical procedures can reduce postoperative complications such as local infections or the more serious "infective endocarditis" [11,12].

Changes in the taxonomic composition of the microbiome are essential when determining a case of periodontitis [13,14]. On other hand Nędzi-Góra etal found that periodontitis grade may not be distinguished according to microbial analysis of subgingival biofilm [15]. Periodontitis is a mixed infection, and, hence, no single antimicrobial inhibits all periodontal pathogens. Thus, the combination of two or more antimicrobials represents a reasonable approach in periodontal therapy (when indicated). Among the recommended combination therapies are metronidazole–amoxicillin for periodontal infections [16].

Mounting evidence exists in support of using of an adjunctive antibacterial to give a more positive clinical response than mechanical therapy alone for the treatment of refractory, aggressive periodontitis and acute necrotizing ulcerative periodontitis (ANUP) [3,17]. Furthermore, antimicrobials have been shown to play a significant role in controlling what was previously categorized as aggressive periodontitis. The regimens of Metronidazole combined with Amoxicillin or Ciprofloxacin and Clindamycin are effective and are preferable to regimens containing Doxycycline.

Azithromycin has been reported as a valid alternative to the regimen of Amoxicillin combined with Metronidazole [11]. Two meta-analyses concluded that systemic antimicrobials use as an adjunctive to scaling and root planning (SRP) significantly improve the clinical outcomes [18,19]. With the emergence of the latest periodontal classification system [20], the previous classification for periodontal diseases, including chronic, aggressive, and some forms of rapidly progressive periodontitis, are now classified under "periodontitis" with a grade and stage system.

SRP is the standard and conventional approach for non-surgical periodontal therapy. However, in cases where a localized recurrent and/or residual pocket depth (PD) of $\geq 5$ mm with inflammation is still present, local drug delivery should be considered, although it might not be as good a choice in cases where multiple sites with pocket depth $\geq 5$ mm exist in the same quadrant, there is a presence of anatomical defects, and it fails to reduce the pocket depth [21].

Currently, SRP plus adjunctive local therapy could potentially be considered a new standard for non-surgical periodontal therapy [16]. Food and Drug Administration (FDA) labels provide complete information for the proper use of locally delivered, controlled-release antimicrobials [16].

Basically, local drug delivery is said to have fewer side effects and reduces or eliminates the possibility of antimicrobial resistance development [22].

Most studies evaluating the knowledge about local and systemic administration of antimicrobials in dental practice focused, in general, on pulpal and apical infections [23–25]. A few other studies evaluated the knowledge of students and dentists about a few aspects of using antimicrobials in periodontal therapy [9,26,27]. Therefore, this study aimed to evaluate the knowledge and practices among dental students and dental practitioners in Saudi Arabia regarding antimicrobial prescription in the context of periodontal therapy and to assess the potential factors that might affect their knowledge and practice.

## 2. Results

Table 1 presents the demographic data of the participants. A total of 223 participants responded to this survey; 148 (66.4%) of them were male. About 55% of them were aged between 25 and 40. Over half of the participants were 6th-year students or interns (27.8%

and 24.2%, respectively); general dental practitioners (GDP) and periodontists represented 13% and 16%, respectively. Most of the respondents (49%) reported that they see more than 15 patients per week, and 66.4% of them reported working in the public sector. Sixty-five percent of the participants reported less than 5 years of experience.

**Table 1.** Characteristics of the study sample (*n* = 223).

| Variable | Number | % |
|---|---|---|
| **Gender** | | |
| Male | 148 | 66.4 |
| Female | 75 | 33.6 |
| **Age** | | |
| <25 years | 54 | 24.2 |
| 25–40 years | 123 | 55.2 |
| >40 years | 46 | 20.6 |
| **Qualifications** | | |
| GDP | 29 | 13.0 |
| 6th-year students | 62 | 27.8 |
| Interns | 54 | 24.2 |
| Periodontists | 36 | 16.1 |
| Other specialties | 42 | 18.8 |
| **Duration of Practice after Graduation** | | |
| <5 years | 145 | 65.0 |
| 5–10 years | 21 | 9.4 |
| >10 years | 57 | 25.6 |
| **Number of Patients/Week** | | |
| <5/week | 52 | 23.3 |
| 5–15/week | 62 | 27.8 |
| >15/week | 109 | 48.9 |
| **Place of Work** | | |
| Public | 148 | 66.4 |
| Private | 75 | 33.6 |

Only around 35% of study participants reported attending courses or continuing education programs on the use of antimicrobials in periodontal therapy within the previous 2 years (Supplementary Table S1). Table 2 presents the participants' responses to antimicrobial use for different periodontal situations. Periodontal abscess and ANUP were the situations for which the participants prescribed antimicrobials most frequently (84.3% each), followed by post-periodontal surgery (79%) and aggressive periodontitis (68%).

Few participants reported prescribing antimicrobials for generalized gingival recession (10.8%) and tooth mobility and furcation involvement (12.1% each).

Surprisingly, 30.9% and 30.5% of the participants reported prescribing systemic antimicrobials for the treatment of acute gingivitis associated with herpes simplex in children and a deep localized periodontal pocket, respectively.

Participants from the private and public sectors showed almost similar responses regarding the prescription of antimicrobials for treatment of different signs, symptoms, and periodontal diseases, except for chronic periodontitis (Periodontitis Grade B), for which participants in the private sector reported more frequent prescription than participants in the public sector (30.7% versus 10.1%; $p < 0.01$; Supplementary Table S2). In contrast to periodontists, participants from other specialties, followed by GDP, and to a lesser extent senior students and interns, reported more frequent prescriptions of antimicrobials for gingival enlargement (5.6% versus 23.8%, 34.5%, 29%, and 35.2%, respectively; $p = 0.02$), periodontitis grade A (0% versus 28.6%, 3.1%, 14.5%, and 13%, respectively; $p = 0.002$),

periodontitis grade B (5.6% versus 28.6%, 13.8%, 22.6%, and 11.1%, respectively; $p = 0.04$), gingival tumors (5.6% versus 26.2%, 31%, 17.7%, and 29.6%, respectively; $p = 0.04$), and post-periodontal surgery (58.3% versus 88.1%, 79.3%, 80.6%, and 89%, respectively; $p = 0.008$).

**Table 2.** Indications for which participants used antimicrobials ("Yes" responses; $n = 223$).

| Indications | Number | % |
|---|---|---|
| Severe pain | 48 | 21.5 |
| Gingival bleeding | 28 | 12.6 |
| Gingival enlargement | 59 | 26.5 |
| Generalized gingival recession | 24 | 10.8 |
| Deep localized periodontal pocket | 68 | 30.5 |
| Tooth/teeth mobility | 27 | 12.1 |
| Periodontal abscess | 188 | 84.3 |
| Acute gingivitis associated with herpes simplex in children | 69 | 30.9 |
| Furcation involvement | 27 | 12.1 |
| Chronic periodontitis (Grade A according to new classification) | 29 | 13 |
| Chronic periodontitis (Grade B according to new classification) | 38 | 17 |
| Aggressive periodontitis (Grade C according to new classification) | 152 | 68.2 |
| Gingival tumors | 49 | 22 |
| Refractory periodontitis | 97 | 43.5 |
| Post-periodontal surgery | 178 | 79.8 |
| Acute necrotizing periodontal disease | 188 | 84.3 |
| In maintenance period after finishing treatment | 32 | 14.3 |

The opposite applies to periodontitis grade C (91.7% versus 78.8%, 31%, 66.1%, and 70.4%, respectively; $p < 0.001$) and refractory periodontitis (80.6% versus 47.6%, 20.7%, 30.6%, and 24.6%, respectively; $p < 0.001$).

Around 90% of participants reported prescribing antimicrobials for AUNP, except for GDP (51.7%) and periodontists (83.3%, $p < 0.001$; Supplementary Table S3). In the context of the type of antimicrobial prescribed for specific periodontal scenarios (aggressive periodontitis, necrotizing ulcerative periodontitis, and periodontal abscess), Table 3 shows that Metronidazole (72.6%, 67.7%, and 61.4%, respectively), Amoxicillin/Clavulanic acid (63.2%, 59.2%, and 72.7%, respectively) and Amoxicillin (59.6%, 54.7%, and 69.5%, respectively) were reported as the most frequently prescribed antimicrobials. These antimicrobials, in the above-mentioned scenarios, were statistically reported more frequently by periodontists, followed by senior students and interns.

In contrast, Spiramycin and Minocycline were reported as being prescribed more frequently by GDP and other specialists. Moreover, more GDP reported not prescribing antimicrobials at all in cases of aggressive periodontitis and necrotizing ulcerative periodontitis (Supplementary Table S4).

**Table 3.** Antimicrobials used for different periodontal diseases ("Yes" responses; *n* = 223).

| Disease | Antibiotic | Number | % |
|---|---|---|---|
| Aggressive periodontitis | Amoxicillin | 133 | 59.6 |
| | Augmentin | 141 | 63.2 |
| | Spiramycin | 33 | 14.8 |
| | Azithromycin | 40 | 17.9 |
| | Clindamycin | 94 | 42.2 |
| | Tetracycline | 48 | 21.5 |
| | Doxycycline | 76 | 34.1 |
| | Metronidazole | 162 | 72.6 |
| | Ciprofloxacin | 41 | 18.4 |
| | Minocycline | 37 | 16.6 |
| | Erythromycin | 33 | 14.8 |
| | Combination of two antibiotics | 46 | 20.6 |
| Necrotizing ulcerative | Amoxicillin | 122 | 54.7 |
| | Augmentin | 132 | 59.2 |
| | Spiramycin | 34 | 15.2 |
| | Azithromycin | 38 | 17 |
| | Clindamycin | 68 | 30.4 |
| | Tetracycline | 42 | 18.8 |
| | Doxycycline | 52 | 23.3 |
| | Metronidazole | 151 | 67.7 |
| | Ciprofloxacin | 35 | 15.7 |
| | Minocycline | 25 | 11.2 |
| | Erythromycin | 29 | 13 |
| | Combination of two antibiotics | 43 | 19.3 |
| Periodontal abscess | Amoxicillin | 155 | 69.5 |
| | Augmentin | 162 | 72.6 |
| | Spiramycin | 26 | 11.7 |
| | Azithromycin | 28 | 12.6 |
| | Clindamycin | 87 | 39 |
| | Tetracycline | 24 | 10.8 |
| | Doxycycline | 32 | 14.3 |
| | Metronidazole | 137 | 61.4 |
| | Ciprofloxacin | 29 | 13 |
| | Minocycline | 17 | 7.6 |
| | Erythromycin | 24 | 10.8 |
| | Combination of two antibiotics | 32 | 14.3 |

Similarly, in the context of the above-mentioned scenarios, participants from the public sector reported more frequently prescribing Amoxicillin and Minocycline, while participants from the private sector reported more frequently prescribing Spiramycin and Tetracycline (Supplementary Table S5). Prescribing a combination of two systemic antimicrobials in the context of periodontal diseases was reported by 81% of participants: 82% preferred to combine Amoxicillin with Metronidazole, and 61.3% preferred to combine Augmentin with Metronidazole (Table 4). Regarding the antibiotic of choice prescribed for patients who are allergic to Penicillin, the majority of participants prescribed Clindamycin (72.7%) followed by Azithromycin (39%), Metronidazole (33.6%), and Erythromycin (28.3%; Table 5).

**Table 4.** Combination of antimicrobials in treatment of periodontal diseases ("Yes" responses).

| Antibiotics | Number | % |
|---|---|---|
| Do you combine more than one antibiotic in the treatment of some periodontal diseases? (*n* = 222) | 180 | 81.1 |
| (1) Amoxicillin with metronidazole (*n* = 196) | 161 | 82.1 |
| (2) Spiramycin with metronidazole (*n* = 189) | 28 | 14.8 |
| (3) Tetracycline with metronidazole (*n* = 187) | 15 | 8 |
| (4) Augmentin with metronidazole (*n* = 194) | 119 | 61.3 |
| (5) Ciprofloxacin with metronidazole (*n* = 189) | 35 | 18.5 |
| (6) Azithromycin with metronidazole (*n* = 190) | 23 | 12.1 |

**Table 5.** Antimicrobials prescribed for penicillin-sensitive patients ("Yes" responses; *n* = 223).

| Antibiotics | Number | % |
|---|---|---|
| (1) Spiramycin | 24 | 10.8 |
| (2) Azithromycin | 87 | 39 |
| (3) Clindamycin | 162 | 72.7 |
| (4) Tetracycline | 25 | 11.2 |
| (5) Doxycycline | 44 | 19.7 |
| (6) Metronidazole | 75 | 33.6 |
| (7) Ciprofloxacin | 41 | 18.4 |
| (8) Minocycline | 14 | 6.3 |
| (9) Erythromycin | 63 | 28.3 |

As shown in Table 6, around 66% of participants reported that mechanical periodontal treatment alone, without any adjunctive antimicrobial therapy, is adequate to resolve the clinical condition in most cases of periodontal disease; the highest proportions were from the periodontists category. Almost half (51.1%) of participants reported that they prescribe antimicrobials (when indicated) for one week, and 31.8% of them reported doing so for 3–5 days. The prescription of systemic antimicrobials after surgical periodontal therapy was reported by 40.4%; the highest proportion was from the periodontists category. Antimicrobials use before and after surgical periodontal therapy was reported by 35.4% of participants. Interestingly, 16.1% of participants reported that there was no need for an antimicrobial prescription along with surgical periodontal therapy; the highest proportion was from the periodontists category. Almost 69.5% of participants reported that there was no need to prescribe Doxycycline hyclate (Periostat) as a sub-antimicrobial dose to treat periodontal diseases; the highest proportion was from the periodontists category. Around 45% of participants reported their agreement regarding the local use of antimicrobials in the context of periodontal therapy. The following are the most frequent indications: a recurrent periodontal pocket (45.4%), a periodontal pocket (45.3%), refractory periodontitis (43.7%), and a periodontal abscess (34.4%; Table 7); the highest proportion was from the periodontists category. Most participants agreed that a lack of knowledge and a lack of postgraduate training are among the barriers for not prescribing antimicrobials in the context of periodontal therapy (64.1% each). Around 35.5% agreed that high cost is a barrier. Around 23.3% agreed that a local antimicrobial is not needed, and 23.8% reported an unsuccessful previous use of it. Around 35.4% thought that a lack of supporting research data is a barrier (Table 8).

**Table 6.** Mechanical and surgical therapy, duration of use of antimicrobials and their potential side effects, and use of sub-antimicrobial doses in the context of periodontal therapy (*n* = 223).

| Question | *n* | % |
|---|---|---|
| Do you think that mechanical periodontal treatment alone without adjunctive antimicrobial therapy is adequate to resolve the clinical condition in most cases of periodontal diseases? | | |
| Yes | 147 | 65.9 |
| No | 76 | 34.1 |
| What is the duration of using systemic antibiotics usually you prescribe for periodontal treatment? | | |
| 3–5 days | 71 | 31.8 |
| 1 week | 114 | 51.1 |
| 2–3 weeks | 18 | 8.1 |
| 1 month | 3 | 1.3 |
| When do you think systemic antibiotics is prescribing in surgical periodontal therapy? | | |
| Before surgery | 18 | 8.1 |
| After surgery | 90 | 40.4 |
| Before and after surgery | 79 | 35.4 |
| No need | 36 | 16.1 |
| What do you think the most important complications which result from improper using systemic | | |
| Toxicity | 21 | 9.4 |
| Diarrhea | 7 | 3.1 |
| Antibiotic resistance | 189 | 84.8 |
| Fungal Infection | 6 | 2.7 |
| Do you prescribe doxycycline hyclate (Periostat) as sub-antimicrobial dose to treat periodontal disease? | | |
| Yes | 68 | 30.5 |
| No | 155 | 69.5 |

**Table 7.** Use of local antibiotics ("Yes" responses).

| Indications | Number | % |
|---|---|---|
| Do you apply local antibiotic in the treatment of periodontal diseases? (*n* = 218) | 97 | 44.5 |
| (1) Periodontal abscess (*n* = 180) | 62 | 34.4 |
| (2) Gingival recession (*n* = 180) | 28 | 15.6 |
| (3) Periodontal pocket (*n* = 180) | **82** | 45.3 |
| (4) Furcation involvement (*n* = 181) | 42 | 23.3 |
| (5) Gingival enlargement (*n* = 179) | 26 | 14.5 |
| (6) Recurrent periodontal pocket (*n* = 183) | 83 | 45.4 |
| (7) Refractory periodontitis (*n* = 183) | 80 | 43.7 |

**Table 8.** Barriers of not using local antibiotics (*n* = 223).

| Barriers | Number | % |
|---|---|---|
| **(1) Not Enough Knowledge about Local Antibiotics** | | |
| Agree | 143 | 64.1 |
| Disagree | 39 | 17.5 |
| Not sure | 41 | 18.4 |
| **(2) High Cost of Local Antibiotics** | | |
| Agree | 80 | 35.9 |
| Disagree | 64 | 28.7 |
| Not sure | 79 | 35.4 |
| **(3) Not Needed** | | |
| Agree | 52 | 23.3 |
| Disagree | 104 | 46.6 |
| Not sure | 67 | 30.0 |
| **(4) Lack of Supporting Research Data** | | |
| Agree | 79 | 35.4 |
| Disagree | 63 | 28.3 |
| Not sure | 81 | 36.3 |
| **(5) Lack of Postgraduate Training** | | |
| Agree | 143 | 64.1 |
| Disagree | 27 | 12.1 |
| Not sure | 53 | 23.8 |
| **(6) Unsuccessful Previous Usage** | | |
| Agree | 53 | 23.8 |
| Disagree | 70 | 31.4 |
| Not sure | 100 | 44.8 |

## 3. Discussion

In general, this study shows the inappropriate practice of dental professionals regarding the prescription of systemic antimicrobials in the context of periodontal disease. Most participants (84.3%) reported prescribing antimicrobials for the treatment of a periodontal abscess. A comparable result (88.1%) was obtained by Naveen et al. [26]. However, this is not an appropriate practice. Basically, antimicrobials are indicated for the treatment of a periodontal abscess when there are signs and symptoms of systemic involvement. Instead, drainage and periodontal debridement of the periodontal abscess mostly resolve the condition without the need for antimicrobial therapy [22]. In our study, apart from periodontists who reported more frequently prescribing antimicrobials in the treatment of refractory and aggressive periodontitis (80.6% and 91.7%, respectively), most participants in other categories reported the opposite (Supplementary Table S3). In general, the prescription of systemic antimicrobials for the treatment of chronic periodontitis was low (13%) in our study compared to the Naveen et al. [26] and AboAlSamh et al. studies [27] (47.5% and 44.25%, respectively).

The use of systemic antimicrobials in the treatment of ANUP and aggressive periodontitis was reported less frequently by GDP compared to other groups. However, with time, clinicians have realized that some patients, despite ideal maintenance, continue to experience periodontal destruction. Considerable fractions of the study's participants (except periodontists) reported prescribing antimicrobials in the context of gingival enlargement/tumors, a deep localized periodontal pocket, and acute gingivitis associated with herpes simplex in children. However, the evidence-based practice is against the prescription of antimicrobials for the above-mentioned scenarios [3,17,28,29].

In fact, promising outcomes were reported using a combination of Amoxicillin plus Metronidazole in the treatment of advanced cases of periodontitis [3,17,30,31], which is con-

sistent with the results of our study (82%). The second preferable antimicrobial combination in our study was Amoxicillin/Clavulanic acid with Metronidazole (61.3%). However, there are no documented studies that support the use of a combination of Amoxicillin/Clavulanic acid with metronidazole for periodontal therapy. This is, probably, because Amoxicillin is less expensive than Amoxicillin/Clavulanic acid and is safer on the stomach [3]. Moreover, Clavulanic acid can cause cholestatic hepatitis, hepatobiliary disorders, hepatotoxicity, and severe adverse events in older patients [17]. Worth noting is that 18.9% of the "other specialties" group reported prescribing a combination of Tetracycline with Metronidazole, although this combination is contraindicated [32].

GDP and dental specialists, other than periodontists, had a lower knowledge about the prescription of antimicrobials in the context of periodontal therapy. Regarding the prescription of the common recommended antibiotics (Metronidazole, Amoxicillin and Augmentin) by the different professionals, periodontists reported as prescribing these more frequently, followed by the 6th-year students and interns (Supplementary Table S4).

GDP prescribes antimicrobials for the indicated periodontal diseases (ANUP, periodontitis grade C and refractory periodontitis) the least of the dental professionals. However, the prescription percentage of 6th-year students for the same clinical problems was high, nearly around the percentages of periodontists. This can be explained by the fact that GDP did not receive ongoing education and that their patients were treated empirically, while sixth-year students were still enrolled in school and dedicated to following scientific methods.

To be successful with antimicrobial periodontal therapy, the periodontal disease condition and the antimicrobial regimen must be properly assessed. Additionally, knowledge about current guidelines and protocols for antibiotic prescription in the periodontal disease context is mandatory [3]. Hence, the referral of periodontal cases to, or at least treating them with a close arrangement with a periodontist, is a must.

It seems that most participants in our survey, except for periodontists, did not have enough information about using a sub-antimicrobial regimen for host modulation in the treatment of periodontal disease; 61.1% of periodontists reported prescribing Doxycycline Hyclate (Periostat) as a sub-antimicrobial dose, in contrast to less than 28% by the dental professionals in all other categories. In fact, studies recommend the use of a sub-antimicrobial dose in combination with mechanical periodontal therapy, as this improves the clinical periodontal parameters and biomarker levels in the gingival crevicular fluid of periodontitis patients [33,34].

In our study, around 44.5% of participants reported their support of using a local antimicrobial therapy in the treatment of recurrent periodontal diseases, namely, periodontal pockets and refractory periodontitis, a result that is higher than what was reported by Choudhury et al. [35] (8.9%). However, there is no sound evidence regarding the effectiveness of using local antimicrobials in the treatment of refractory periodontitis. In our study, all categories of dental professionals, except periodontists, reported supporting local antimicrobials use when they are not indicated, namely, for a periodontal abscess, periodontal pockets, recurrent pockets, and refractory periodontitis [36]. Nevertheless, the cost-effectiveness evaluation of local antimicrobials use is still controversial [37]. The following were reported as barriers against the use of local antimicrobials: not enough knowledge, a lack of local postgraduate training, the high cost, and a lack of supporting research data. The latter two are consistent with the results of other studies [35,37].

The provision of a continuing dental education program offers an important resource that helps to develop and update knowledge about the use and misuse of antimicrobials, which in turn will have an effect on prescribing practices. In addition, following the guidelines that are set by different professional organizations is of the utmost importance. We recommend, therefore, to organize several training programs and scientific meetings with students, GDP, and dental specialists, other than periodontists, to increase their awareness and knowledge about periodontal therapy and to guide them in their selection of the most appropriate antimicrobials.

*Study Limitations*

This study has some limitations. First, it was a descriptive study, so the design is at a low level on the evidence pyramid. Second, the tool of assessing the outcomes of interest was a questionnaire, a tool that is criticized due to many aspects including, among others, dishonest responses, variables' understanding, long questionnaires being boring, and recall basis. In studies on health practices, it is more appropriate to record the real practice; in the case of our study, these were the prescriptions from the archives of the centers where the participants work. Third, the sample size was relatively small, although we sent many reminders, with the aim of increasing the number of participants; as the increment was very low after the last reminder, we stopped. Hence, the results cannot be generalized to dental practitioners who are working in Saudi Arabia. Fourth, the participants were unequally distributed, as the majority of the study participants were males. Fifth, the available guidelines on antimicrobial use in the context of periodontal therapy are somewhat inconsistent, as is the case with most health practices. However, we contrasted our results with the best evidence available. In order to overcome the above-mentioned shortcomings, and, hence, to obtain a more reliable result, large-scale well-designed studies are highly encouraged.

## 4. Materials and Methods

**Study Design and Settings:** This was a descriptive study using an anonymized online questionnaire. The study was conducted in 2020/2021 academic year as a part of the internship projects at College of Dentistry, Jazan University, Saudi Arabia. The study was approved by the Scientific Research Unit, College of Dentistry, Jazan University (Protocol No. CODJU-20221).

**Sample Selection**: The study targeted as many senior dental students, dental interns, general dental practitioners (GDP) and dental specialists, who were working that time in private or public dental college/centers and hospitals in Saudi Arabia, as possible. To approach the potential sample, a questionnaire (see below) was sent online by email or on social media. In order to be specific, the authors sent the link to many dental professional groups that they were currently joining via social media. Further, the authors asked the members of these groups to share the link with their colleagues. The link included, besides the questionnaire and an invitation to participate, a message introducing the purpose of the study and assuring anonymity of the survey. Only those who chose "Agree to participate" were allowed to complete the questionnaire.

**Questionnaire:** The questionnaire was adopted from a previous study with modifications [35]. It consisted of three parts. The first part covered demographic data (age, gender, qualification/specialty (senior students, interns GDP, periodontists, and other specialties)), duration of practice after graduation, number of patients per week, and place of work (public or private). The second part covered the following: whether the participants attended any course or continuing education program on use of antimicrobials; the source of information about antimicrobial prescription; and use of systemic antimicrobial therapy in the treatment of periodontal diseases (antimicrobials prescription for various clinical periodontal signs and/or symptoms). The following clinical situations were included: severe pain, gingival bleeding, gingival enlargement, generalized gingival recession, deep localized periodontal pocket, tooth mobility, periodontal abscess, acute gingivitis associated with herpes simplex in children, chronic periodontitis (Grade A and Grade B according to the new classification), aggressive periodontitis (Grade C according to the new classification), gingival tumors, refractory periodontitis, post-periodontal surgery, acute necrotizing periodontal disease (ANUP), and for maintenance therapy.

The preferred antimicrobial (Amoxicillin, Augmentin, Spiramycin, Azithromycin, Clindamycin, Tetracycline, Doxycycline, Metronidazole, Ciprofloxacin, Minocycline, or Erythromycin) and duration of use were also reported in the context of treatment of aggressive periodontitis, necrotizing periodontal disease, and periodontal abscess. An alternative antimicrobial in case of Penicillin allergy was reported too. The participants were also asked

whether they use antimicrobials in association with mechanical therapy or as monotherapy. The participants were asked about the timing of antimicrobial use (before, before and after, or only after the surgical procedure). Knowledge about the potential complications of systemic antimicrobials was assessed. Apart from the systemic antimicrobials, the use of Doxycycline hyclate (Periostat) as sub-antimicrobial dose to treat periodontal disease was reported. The following situations were included: periodontal abscess, gingival recession, periodontal pocket, furcation involvement, gingival enlargement, recurrent periodontal pocket, and refractory periodontitis, along with the possible reasons for not using local antimicrobials in periodontal therapy.

**Statistical analyses:** Analyses were performed using statistical package for the social sciences (SPSS) program, Version 21 (Armonk, NY, USA: IBM Corp). A *p* value of less than 0.05 was considered significant. A chi-square test was used to assess any associations between the use of antimicrobials and the different variables qualification (GDP, 6th-year students, interns, periodontists, and other specialties) and place of work (public and private).

### 5. Conclusions

This study showed the inappropriate knowledge and practices of dental professionals regarding the prescription of antimicrobials in the context of periodontal disease, except for the periodontists. Prescribing practices can be improved by raising dental professionals' understanding of the proper guidelines and recommended standards for prescribing antimicrobials. Providing continuous, periodic educational programs to improve the practice of antimicrobials use by dental practitioners in the context of periodontal therapy.

**Supplementary Materials:** The following are available online: https://www.mdpi.com/article/10.3390/pharma2010007/s1, Supplementary Table S1: Attendance of continuing education program and source of information on use of antimicrobials in periodontal therapy; Supplementary Table S2: Responses regarding indications of antimicrobials for various clinical symptoms and diagnoses by workplace; Supplementary Table S3: Responses regarding indications of antimicrobials for various clinical signs/symptoms and diagnoses by specialty; Supplementary Table S4: Responses regarding the preferred antimicrobials prescribed for various clinical diagnoses by specialty; Supplementary Table S5: Responses regarding the preferred antimicrobials prescribed for various clinical diagnoses by workplace.

**Author Contributions:** Conceptualization, M.S.A.-A. and E.H.; methodology, M.A.K.; validation, M.S.A.-A., A.M. (Abdulaziz Mahnashi) and J.K.; formal analysis, E.H.; investigation, M.A.K., A.M. (Abdullah Mahnashi) and K.M.; writing—original draft preparation, M.S.A.-A.; writing—review and editing, W.H.A.; supervision, M.S.A.-A.; project administration, M.S.A.-A.; N.F. statistic and revision. All authors have read and agreed to the published version of the manuscript.

**Funding:** This research received no external funding.

**Institutional Review Board Statement:** The study was approved by the Scientific Research Unit, College of Dentistry, Jazan University (Protocol No. CODJU-20221).

**Informed Consent Statement:** The survey was prefaced with an inquiry to participate. Only those who agreed to participate were allowed to go to the survey.

**Acknowledgments:** The authors would like to thank the Saudi Society of Periodontology for helping in the distribution of the questionnaire and all the dental practitioners who kindly completed and returned the questionnaire.

**Conflicts of Interest:** The authors declare no conflict of interest.

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
