# Peer review of "Knowledge and Practices among Dental Practitioners Regarding the Use of Antimicrobials for Periodontal Diseases: An Online Survey in Saudi Arabia"

_2813-0618, doi:10.3390/pharma2010007_

Round 1

Reviewer 1 Report

Thank you for giving me the opportunity to review the manuscript titled “Knowledge and practice of dental practitioners on use of antimicrobials for periodontal diseases: An online survey in Saudi Arabia” This study using online survey revealed that knowledge on and practice of antimicrobials use for periodontal diseases were inadequate especially among practitioners other than periodontists in Saudi Arabia. While this is an extremely interesting new initiative, there are several major concerns that must be addressed before this manuscript can be considered suitable for publication.

1) In Introduction section, the first paragraph is very long, so I think it would be easier for the reader to understand if I put a line break.

2) I think you can provide more specific information about #12 reference.

3) Since the #13 and #14 references are little old, the authors would be better to quote the most recent review, if they can.

4) The authors mentioned, “Little has been known so far concerning knowledge of dental students and dental”, but they provide some references. Therefore, I recommend they revise the manuscript.

5) The Result section, the second paragraph is too long. The authors should revise the manuscript to understand easily for readers.

6) In the first paragraph of the Discussion section, there were sentences that should be included in the Introduction section. This paragraph must briefly state the result of this study and discuss them.

7) The authors should mention about “severe adverse events.” (P8L6)

8) The Discussion section, the second paragraph is so long. The authors should reconsider your paragraph structure.

9) Generalization of the results of this study should be mentioned in the limitation.

10) The manuscript of statistical analysis was inadequate. Please describe how you grouped the participants.

Author Response

Response to Reviewer 1

Comments

This study using online survey revealed that knowledge on and practice of antimicrobials use for periodontal diseases were inadequate especially among practitioners other than periodontists in Saudi Arabia. While this is an extremely interesting new initiative, there are several major concerns that must be addressed before this manuscript can be considered suitable for publication.

Point 1: In Introduction section, the first paragraph is very long, so I think it would be easier for the reader to understand if I put a line break.

Response 1:  As the reviewer suggested, we have added line breaks.

Point 2: I think you can provide more specific information about #12 reference.

Response 2: As the reviewer suggested, we have added more specific information to the #12 reference. It was changed to the #16 reference. 

Point 3:  Since the #13 and #14 references are little old, the authors would be better to quote the most recent review, if they can.

Response 3: As the reviewer suggested, we have replaced the #13 and #14 references with recent references. They were changed to the #18 and # 19 references.

Point 4: The authors mentioned, “Little has been known so far concerning knowledge of dental students and dental”, but they provide some references. Therefore, I recommend they revise the manuscript.

Response 4: As the reviewer suggested, we have revised and modified the statement (in the last part of the introduction).

Point 5: The Result section, the second paragraph is too long. The authors should revise the manuscript to understand easily for readers.

Response 5: As the reviewer suggested, we have revised, shortened, and modified the second part of result to be understood easily by the reader. 

Point 6: In the first paragraph of the discussion section, there were sentences that should be included in the Introduction section. This paragraph must briefly state the result of this study and discuss them.

Response 6: As the reviewer suggested, we have moved the statements that were not related to the results from the first paragraph of the discussion to the introduction part. 

Point 7:  The authors should mention about “severe adverse events.” (P8L6)

Response 7: As the reviewer suggested, we have mentioned about severe adverse events (at the end of the 1st part of the discussion).

Point 8: The Discussion section, the second paragraph is so long. The authors should reconsider your paragraph structure.

Response 8: As the reviewer suggested, we have revised, shortened, and modified the second part of the discussion to be understoodd easily by the reader.

Point 9: Generalization of the results of this study should be mentioned in the limitation.

Response 9: As the reviewer suggested, we explicitly mentioned that the results can’t be generalized (at the end part of the discussion).

Point 10: The manuscript of statistical analysis was inadequate. Please describe how you grouped the participants.

Response 10: As the reviewer suggested, we have detailed the grouping factors by which the participants were distributed (in the part on statistical analysis).

Thank you.

The corresponding author.

Reviewer 2 Report

The authors conducted an online survey in Saudi Arabia on dentists' use and knowledge of antimicrobials for periodontal disease.

Unfortunately, there are major problems with this paper. The research design is inappropriate. The authors stated in the purpose of the study section, "to assess the potential factors that might affect their knowledge and practice." However, it is impossible to investigate the relevant factors for this research theme with the results of a cross-sectional survey and a simple statistical analysis.

No matter how much background and discussion is written, if the objectives and methodology do not match, it has no value as a scientific paper.

And even if the authors quit to consider the factors, a survey that merely asks about usage has no value as a paper.

Thank you very much.

Author Response

Response to Reviewer 2

 Comments

The authors conducted an online survey in Saudi Arabia on dentists' use and knowledge of antimicrobials for periodontal disease.

Unfortunately, there are major problems with this paper.

 Point 1: The research design is inappropriate.

Response 1: We agree with the reviewer. But we confirm that we had mentioned that explicitly as one of the limitations of the study: “This study has some limitations. First, it was a cross-sectional study, the design that has a low level in the evidence pyramid. Second, the tool for assessing the outcomes of interest was a questionnaire, the tool that is criticized due to many aspects including among others dishonest responses, variability in understanding, being boring in case of the long questionnaire, and recall basis. In studies on health practice, it is more appropriate to record the real practice: in the case of our study, prescriptions from the archives of the centers where the participants work.”

 Point 2: The authors stated in the purpose of the study section, "to assess the potential factors that might affect their knowledge and practice." However, it is impossible to investigate the relevant factors for this research theme with the results of a cross-sectional survey and a simple statistical analysis.

Response 2: Although we partially agree with the reviewer that the cross-sectional survey and the simple statistical analysis might not be sufficient tools to assess the potential factors that might affect the participants’ knowledge and practice, the results showed many differences by specialty, qualification, place of work, etc. For example, the results clearly showed that periodontists revealed the best knowledge and practice.

Point 3: No matter how much background and discussion is written, if the objectives and methodology do not match, it has no value as a scientific paper.

Response 3: We respectfully disagree with the reviewer. We would have been appreciative if the reviewer had mentioned explicitly what aspects of the objectives and methodology do not match, so we could explain.

Point 4: And even if the authors quit to consider the factors, a survey that merely asks about usage has no value as a paper.

Response 4: We respectfully disagree with the reviewer. The scientific literature is full of similar studies. Indeed, these studies are essential to stand on the real knowledge and practice of the assessed health problems in order to later on tailor the proper health policy to rectify them.

Thank you.

The corresponding author.

Reviewer 3 Report

I would like to conratulate the Authors for their efforts put into preparing the manuscript „Knowledge and practice of dental practitioners on use of anti- microbials for periodontal diseases: An online survey in Saudi Arabia”. The topic is both interesting and important.

In my personal opinion the manuscript is well-rounded and the outcomes are nicely reported. However, there are some aspects that would benefit from further clarification.

Abstract- please report the number of doctors that filled the survey. It will bring broader scope for interpretation of reported % data.

Introduction- please add section allocated to the characteristics of microbiome associated with periodontitis. Underscore the differences between its composition in different clinical scenario and up-date the references (e.g. doi: 10.23736/S0026-4970.18.04198-5; doi: 10.5114/ceji.2020.101256; doi: 10.1093/femsre/fuac052). 

Materials and Methods / Results: these sections are nicely arranged.

Discussion: This section requires some changes, as it seems a little bit chaotic.

Please rewrite the first paragraph to introduce the most important findings of the survey. Concentrate on main differences between attitudes twoards pharmacotherapy between different professionals. Describe how the antibiotics are prescribed by them and in which clinical scenarios. 

Move fragments of texts that describe general traits of periodontal diseases to the introduction, and do not repeat in this section information already given in the introduction. 

In another paragraph concentrate on the guideliness and rules of pharmacotherapy in periodontology. Pinpoint the differences between recomendation and the practice based on the survey. Hypothesize the reasons behind such differences. 

Concentrate first on systemic antibiotcs, and only then move to locally used antibiotics. 

Limitations: Actually the most serious flaw of this study is relatively small numbers of evaluated surveys and the gross differences between sex proportions (number of women is doubled). As the abovementioned migh have bearings of observed findings, please elaborate on them. 

Give also some recommendations for future studies in this area.

Conclusion: please rewrite the conclusion. I would suggest to use bullet points, since it might help underscore critical findings.

Author Response

Response to Reviewer 3

 Comments

I would like to conratulate the Authors for their efforts put into preparing the manuscript „Knowledge and practice of dental practitioners on use of anti- microbials for periodontal diseases: An online survey in Saudi Arabia”. The topic is both interesting and important.

In my personal opinion the manuscript is well-rounded and the outcomes are nicely reported. However, there are some aspects that would benefit from further clarification.

Point 1: Abstract- please report the number of doctors that filled the survey. It will bring broader scope for interpretation of reported % data.

Response 1:

If the reviewer meant the total number to whom the questionnaire was sent, this cannot be calculated as we provided the questionnaire as a link to many social media groups from which it was shared with others.

But regarding the total number of respondents, it was 223, and we have mentioned it explicitly in the abstract as the reviewer suggested.

Point 2: Introduction- please add section allocated to the characteristics of microbiome associated with periodontitis. Underscore the differences between its composition in different clinical scenario and up-date the references (e.g. doi: 10.23736/S0026-4970.18.04198-5; doi: 10.5114/ceji.2020.101256; doi: 10.1093/femsre/fuac052). 

Response 2:

As the reviewer suggested, we have added a section about the microbiome associated with periodontitis to the introduction, and we have added the references related to it in the references (references number 13, 14, and 15).

Point 3: Materials and Methods / Results: these sections are nicely arranged.

Response 3:

Thank you very much for your consideration.

Point 4: Discussion: This section requires some changes, as it seems a little bit chaotic. Please rewrite the first paragraph to introduce the most important findings of the survey.

Response 4:

As the reviewer suggested, we rewrote the discussion to introduce the most important findings of the survey.

Point 5: Concentrate on main differences between attitudes towards pharmacotherapy between different professionals.

Response 5:

As the reviewer suggested, we made the recommended changes.

Point 6: Describe how the antibiotics are prescribed by them and in which clinical scenarios. 

Response 6:

As the reviewer suggested, we made the recommended changes.

Point 7: Move fragments of texts that describe general traits of periodontal diseases to the introduction, and do not repeat in this section information already given in the introduction. 

Response 7: As the reviewer suggested, we moved the parts of the text that describe the general traits of periodontal diseases to the introduction.

Point 8: In another paragraph concentrate on the guidelines and rules of pharmacotherapy in periodontology.

Response 8:

As the reviewer suggested, we made the recommended changes in the introduction.

Point 9: Pinpoint the differences between recommendation and the practice based on the survey. Hypothesize the reasons behind such differences.

Response 9:

 As the reviewer suggested, we made the recommended changes in the discussion.

Point 10: Concentrate first on systemic antibiotcs, and only then move to locally used antibiotics. 

Response 10:

As the reviewer suggested, we made the recommended changes in the discussion.

Point 11: Limitations: Actually the most serious flaw of this study is relatively small numbers of evaluated surveys and the gross differences between sex proportions (number of women is doubled). As the abovementioned might have bearings of observed findings, please elaborate on them. 

Response 11:

As the reviewer suggested, we made the recommended changes at the end of the discussion.

Point 12: Give also some recommendations for future studies in this area.

Response 12:

As the reviewer suggested, we added some recommendations in the conclusion.

Point 13: Conclusion: please rewrite the conclusion. I would suggest to use bullet points, since it might help underscore critical findings.

Response 13:

As the reviewer suggested, we rewrote the conclusion with recommendations using bullet points.

Thank you.

The corresponding author.

Round 2

Reviewer 1 Report

Thank you  for submit the revised manuscript.

I think that the manuscript was improved.

Author Response

(The authors gave the same response as above.)

Reviewer 2 Report

I have never seen such a low level response. All replies are just paraphrases and not worth reading. I will state it again. This paper is worthless. Thank you very much.

Author Response

(The authors gave the same response as above.)

Reviewer 3 Report

Authors applied all the suggestions hence now the manuscript is ready for further processing.

Author Response

(The authors gave the same response as above.)
